# Nasal vs. oral BREATHing WIn Strategies in healthy individuals during cardiorespiratory Exercise testing (BreathWISE)

Massimo Mapelli [1,2‡]*, Elisabetta Salvioni[1‡], Irene Mattavelli[1], Giulia Grilli [1], Gabriele Zerboni [1], Alessandro Nava[1], Nicolò Capra [1], Arianna Galotta [1], Matteo Biroli [1], Gaia Bellini[1], Mattia Dall'Asta[1], Elisabetta Pasini[1], Antonio De Paola[1], Ludovica Torzolini[1], Nicola Mani[1], Sebastiano Turri[1], Jeness Campodonico[1,2], Piergiuseppe Agostoni[1,2]

**1** Centro Cardiologico Monzino, IRCCS, Milan, Italy, **2** Department of Clinical Sciences and Community Health, Cardiovascular Section, University of Milan, Milan, Italy

‡ Shared first author privilege.
* massimo.mapelli@cardiologicomonzino.it

## Abstract

### Background

Nasal and oral exclusive breathing modes have benefits and drawbacks during sub-maximal exercise. It is less known whether these responses would extend to anaerobic work performed at high intensity. The purpose of this study is to find the most efficient mode of breathing during different phases of a maximal exercise at cardiopulmonary exercise test (CPET).

### Methods

Healthy subjects were recruited to perform 4 maximal CPETs (standard conditions (STD), exclusively nasal breathing (eNAS), exclusively oral breathing (eOR), partial nasal breathing (pNAS) with just one blocked nostril) using the same ramp protocol on an electronically braked cycle ergometer. Before the exercise a standard spirometry was executed in the same order. Twelve healthy subjects ($28.6 \pm 5.2$ y, 50% males) performed the 4 CPETs within one month. Variables were analysed at rest, at anaerobic threshold (AT), at intermediate exercise steps, and at peak.

### Results

Compared to STD, eOR, and pNAS conditions, eNAS was associated with a significant lower peakVO$_2$, peakVCO$_2$, peak ventilation, respiratory rate, VE/VCO$_2$ slope, respiratory exchange ratio, and workload ($p < 0.05$ for all). Moreover, peak inspiration and peak expiration time were augmented, while forced expiratory volume and vital

**Data availability statement:** Repository of raw data available without restrictions at https://zenodo.org/badge/DOI/10.5281/zenodo.14280724.svg.

**Funding:** The author(s) received no specific funding for this work.

**Competing interests:** The authors have declared that no competing interests exist.

capacity at rest were reduced. Only minor differences were detected at rest or AT. eNAS breathing Borg scale was higher in all phases of the exercise.

## Conclusions

In young healthy subjects, an exclusively nasal respiration induces significant impairment on peak exercise capacity at CPET due to ventilatory limitation, with only minor effects on metabolic parameters at rest and in submaximal effort.

---

## Introduction

Respiration is a fundamental physiological process essential for sustaining life and optimizing physical performance. Normally in healthy individuals, breathing predominantly occurs through the nasal cavity at rest, which is highly recommended when fluids sparing is needed as it happens in cold environments or at high altitude [1]. Differently, a combination of nasal and oral breathing is more common during exercise to facilitate greater air exchange [2]. In addition to these standard breathing strategies, there are circumstances where individuals may be required to breathe with one or both nostrils blocked or through the mouth due to external factors such as nasal congestion from a cold or other condition affecting nasal airflow. Various breathing modes— exclusively oral (eOR), esclusively nasal (eNAS), standard oronasal (combination of mouth and nose; STD), and partial nasal breathing with just one nostril blocked (pNAS)—can significantly impact the body's response during exercise. Each breathing mode offers unique advantages and disadvantages during different exercise intensities. Nasal breathing, for instance, helps filter and humidify the air, increase nitric oxide production, and promote efficient oxygen uptake ($VO_2$) and carbon dioxide removal ($VCO_2$) at lower intensities [3–5]. In contrast, oral breathing allows for a higher volume of air to be inhaled and exhaled, which is particularly beneficial during high-intensity exercise when oxygen demand is high and rapid gas exchange is necessary. Few studies have explored the effects of different breathing modes on physiological responses during exercise. Research has shown that oral breathing can produce higher $VO_2$, ventilation (VE), and respiration rates (RR) compared to an exclusively nasal breathing, particularly at moderate to high intensities [6–10]. This suggests that adding oral breathing might be more effective in meeting the increased oxygen demands of the body during intense physical activity. However, the transition from nasal to oral breathing, the specific conditions under which each mode becomes advantageous, and in particular at which exact extent the exercise performance could be affected, are not entirely understood. Previous studies have typically focused on either rest or submaximal intensities. Understanding the preferred breathing mode at various intensities could provide valuable insights for improving athletic performance and respiratory efficiency.

This study aims to compare the physiological effects of different breathing modes (eNAS, pNAS, eOR, STD) during a maximal cardiopulmonary exercise test (CPET). By examining the responses at different phases of maximal exercise, the study seeks

to quantify the respiratory and metabolic differences under the four type of respirations. Such information could help individuals improve their performance by adopting the most suitable breathing strategy for different exercise intensities.

## Methods

This research follows an interventional, prospective, randomized, double-blind, and crossover design. Between 19-06-2024 and 15-09-2024, healthy participants, both male and female, were recruited.

### Ethical statement:

The study was approved by the Ethical Committee of Centro Cardiologico Monzino (Milano, Italy) and registered as R1925/24 – L2-129. The protocol adheres to the Declaration of Helsinki. All patients signed the informed consent before the study procedures.

The inclusion criteria included being 18 years or older. The exclusion criteria included having underlying cardiorespiratory diseases, a history of COVID-19 infection, ongoing chronic medication, pregnancy, or any inability or clinical contraindications to performing maximal exercise. Participants were instructed to avoid intense physical activity for 24 hours before each test. All participants were nonsmokers.

All subjects underwent four consecutive CPETs performed at least 24 hours apart but within one month, in the following conditions: standard conditions (STD), exclusively nasal breathing (eNAS), exclusively oral breathing (eOR), partial nasal breathing (pNAS) with just one blocked nostril) (Fig 1). The absence of lateral air leakage was carefully verified as a standard used procedure in CPET laboratories before each test by completely blocking the ventilation valve of the spirometry mask with the palm of the hand as previously described [11].

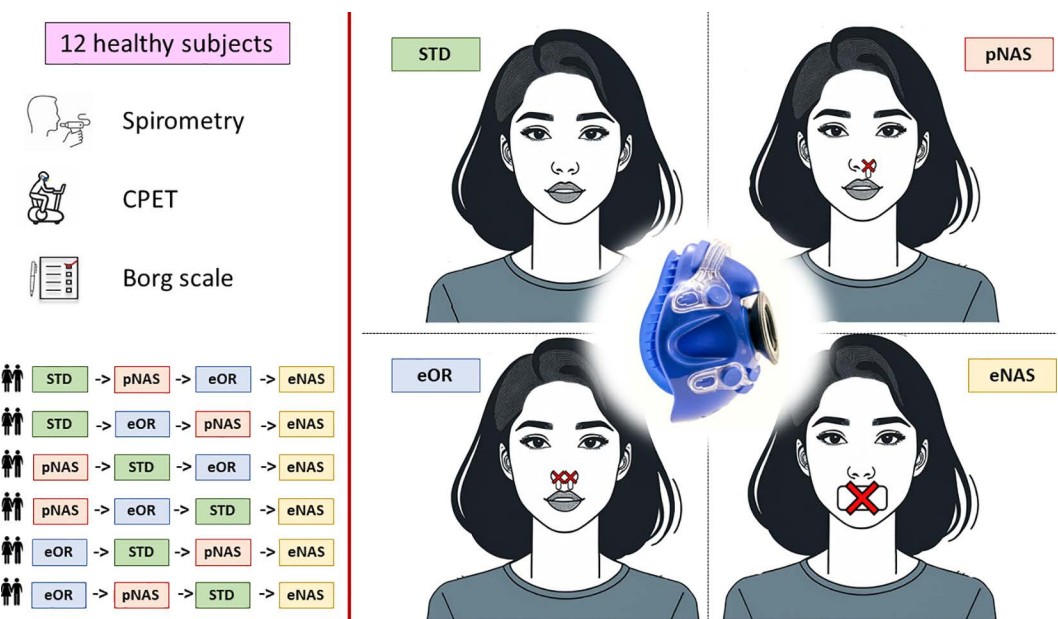

**Fig 1. Study procedures.** Schematic representation of the protocol procedures. The 12 subjects were randomized to perform the tests in the four different conditions: 1) standard conditions (STD), 2) exclusively nasal breathing (eNAS), 3) exclusively oral breathing (eOR), 4) partial nasal breathing (pNAS) with just one blocked nostril using the same ramp protocol on an electronically braked cycle ergometer. Before the exercise a standard spirometry was executed in the same order. Borg scale was collected at rest and during exercise. *Abbreviations*: STD: standard conditions; eNAS: exclusively nasal breathing; eOR: exclusively oral breathing; pNAS: with just one blocked nostril; CPET: cardiopulmonary exercise test.

The execution order of the CPETs was assigned in a randomized fashion to cover all possible combinations. Moreover, tests were executed approximatively at the same time of the day (late afternoon).

## Cardiopulmonary Exercise Testing (CPET)

CPET was performed on an electronically braked cycle ergometer (Corival-Lode, The Netherland) using a personalized ramp protocol set, to reach peak exercise in $10 \pm 2$ minutes [12] and constant in all the 4 evaluations. During the execution of the tests, all the subjects were allowed to see their rpm to maintain a constant pedaling rhythm (60–65 rpm) in all the tests, but all other variables, including time, workload, heart rate (HR) and gas exchange parameters were obscured to them. CPETs were performed and analyzed as standard [13]. Specifically, in the absence of clinical events, tests were self-interrupted by subjects when they reported the maximal effort. Subjects wore a mask to measure VE and respiratory gases breath by breath (Quark PFT Cosmed cart, Roma, Italy). During the test HR and a 12-lead ECG were continuously monitored, Hb $O_2$ saturation ($SpO_2$) was recorded by an oximeter, and blood pressure was monitored with a cuff sphygmomanometer at rest and every two minutes. Peak$VO_2$ was calculated as the 30 seconds average of the highest $VO_2$ recorded, while the VE/$VCO_2$ slope was calculated as the slope of the linear relationship between VE and $VCO_2$ from 1 minute after the start of loaded exercise and the end of the isocapnic buffering period and also expressed as percent of predicted [14]. Predicted peak$VO_2$ was calculated by Hansen and Wasserman equation as (height-age) x 20 for men and (height-age) x 14 for women [15]. Anaerobic threshold (AT) was measured by V-slope analysis of $VO_2$ and $VCO_2$ [16]. $VO_2$/work relationship was measured through the entire exercise protocol. Respiratory exchange ratio (RER) was calculated as the ratio between $VCO_2$ and $VO_2$. A maximal RER during exercise exceeding 1.05 was considered an indicator of a maximal effort [17]. Similarly, a maximal peak HR above 100% was also used as a supportive criterion [17–19]. All tests were analyzed *a posteriori* by a CPET expert blinded to the steps of the study. Specifically, data during exercise were analyzed as follows: six steps of exercise were considered: rest, peak, AT, 25%, 50% and 75% of maximal workload reached by each individual in their test in the STD condition. Consequently, for eNAS, pNAS, and eOR tests, intermediate steps data were reported at the workload (Watt) corresponding to 25, 50 and 75% of maximal workload of the STD test, therefore the four conditions were compared iso-watt in each subject, as previously described [20]. Accordingly, except than at peak exercise, respiratory and gas exchange parameters were analyzed in each patient at iso-Watts. Subjects' degree of dyspnea and fatigue were assessed by Borg Scale [21] at rest, after 3 minutes, after 6 minutes and at peak exercise.

Standard pulmonary function tests were performed in all study conditions through the CPET mask before the exercise (Quark PFT Cosmed, Roma, Italy). Spirometry was performed according to current guidelines [22]. Predicted values are from Quanjer et al. [23].

## Statistical analysis

Continuous variables are described as mean±standard deviation (SD).Categorical variables are expressed as numbers (percentages). Differences between the three protocol conditions were analysed by repeated measures ANOVA. CPET data were analyzed breath by breath except for peak$VO_2$ analysis (averaged 30 seconds). For each subject, we calculated the workload corresponding to 25%, 50%, and 75% of the maximal load reached during the test performed in STD condition, and we compared the corresponding VE, $VO_2$, and $VCO_2$ values between the CPETs in the four conditions at the same workloads. Analyses were carried out with the SAS statistical package v. 9.4 (SAS Institute Inc., Cary, NC, USA), and all tests were 2-sided. $p < 0.05$ was considered as statistically significant.

Sample size determination: With a sample of 12 subjects we planned to identify an effect size of 1.2, with a power of 80%, considering an alpha = 0.05 and an ANOVA for repeated measures (4 measurements).

## Results

Twelve healthy subjects (28.6 ± 5.2 years, 50% males) performed the 4 CPETs within one month. No adverse events occurred and we did not detect any significant variation in HR, blood pressure, and $SpO_2$ at peak exercise. In all conditions, a maximal or nearly maximal effort was reached, as confirmed by RER > 1.05 in all cases and by a maximal HR values close to 100% of the predicted (Table 2). AT was identified in all subjects and in all study conditions. Variables were analyzed at rest, at AT, at peak exercise, and at intermediate iso-watt steps (25%, 50%, and 75% of the maximal load reached during the test performed in STD condition).

Table 1 shows metabolic and ventilatory data at rest and at the AT. In brief, $VO_2$, $VCO_2$, end tidal partial pressure of $CO_2$ ($PetCO_2$), end tidal partial pressure of $O_2$ ($PetO_2$), VE, tidal volume (TV), and respiratory rate (RR) did not change in the 4 different conditions at rest. Notably, at the AT we observed in the eNAS condition a significant reduction in VE, $PetO_2$ and a higher value of $PetCO_2$.

At peak exercise (Table 2), eNAS was associated with a significantly lower peakVO₂ (28.0 ± 5.8 mL/min/kg vs. 33.4 ± 6.3 in STD), peakVCO₂ (2068 ± 729 mL/min vs. 2701 ± 860), peak VE (55.2 ± 18.3 L/min vs 88.2 ± 20.6), peak RR (30.4 ± 10.6 beat/min vs. 42.4 ± 9.1), RER (1.10 ± 0.10 vs 1.22 ± 0.08), PetO₂ (105.2 ± 9.8 vs 115.2 ± 5.1 mmHg), and workload (184 ± 6

**Table 1. CPET values at rest and at anaerobic threshold.**

|  | STD | pNAS | eOR | eNAS | p |
|---|---|---|---|---|---|
| **FEV1 (L)** | 3.98 ± 1.04 | 3.88 ± 0.97 | 3.97 ± 1.00 | 2.64 ± 0.98[+*#] | 0.035 |
| **FEV1%** | 99.92 ± 11.91 | 97.83 ± 12.63 | 99.83 ± 10.50 | 67.33 ± 20.23[+*#] | 0.016 |
| **FVC (L)** | 4.59 ± 1.14 | 4.52 ± 1.14 | 4.55 ± 1.08 | 4.00 ± 1.10[+*#] | <0.001 |
| **FEV1/FVC (%)** | 87 ± 5 | 86 ± 6 | 87 ± 6 | 69 ± 23 | 0.139 |
| **SBP rest (mmHg)** | 121 ± 20 | 116 ± 17 | 110 ± 21[+] | 114 ± 13 | 0.012 |
| **DBP rest (mmHg)** | 71 ± 10 | 72 ± 12 | 70 ± 13 | 73 ± 9 | 0.647 |
| **HR rest (bpm)** | 82 ± 13 | 79 ± 9 | 79 ± 10 | 77 ± 12 | 0.836 |
| **VO₂ rest (mL/min)** | 338 ± 79 | 315 ± 83 | 351 ± 108 | 342 ± 158 | 0.741 |
| **VCO₂ rest (mL/min)** | 295 ± 86 | 297 ± 91 | 301 ± 91 | 279 ± 131 | 0.945 |
| **VE rest (L/min)** | 12.1 ± 2.7 | 11.6 ± 2.1 | 12.3 ± 2.8 | 10.9 ± 4.1 | 0.639 |
| **RR rest** | 16.7 ± 5.2 | 17.1 ± 3.2 | 17.6 ± 5.0 | 16.1 ± 6.4 | 0.801 |
| **TV rest (L)** | 0.80 ± 0.27 | 0.71 ± 0.20 | 0.75 ± 0.19 | 0.81 ± 0.39 | 0.289 |
| **SpO₂ rest (%)** | 97.8 ± 0.8 | 98.0 ± 0.7 | 97.7 ± 0.5 | 98.2 ± 0.8 | 0.373 |
| **PetCO₂ rest (mmHg)** | 32.2 ± 4.5 | 31.3 ± 3.7 | 31.6 ± 3.8 | 33.1 ± 4.3 | 0.180 |
| **PetO₂ rest (mmHg)** | 110.6 ± 6.6 | 110.8 ± 4.6 | 111.1 ± 3.7 | 107.9 ± 6.5 | 0.333 |
| **VO₂AT (mL/min)** | 1325 ± 534 | 1316 ± 559 | 1366 ± 586 | 1288 ± 637 | 0.772 |
| **VO₂AT (mL/Kg/min)** | 19.73 ± 5.69 | 19.55 ± 5.31 | 20.33 ± 6.12 | 19.21 ± 6.20 | 0.658 |
| **VO₂AT%** | 0.53 ± 0.13 | 0.52 ± 0.12 | 0.55 ± 0.15 | 0.52 ± 0.15 | 0.719 |
| **HR AT (bpm)** | 137 ± 24 | 135 ± 19 | 135 ± 16 | 132 ± 15 | 0.938 |
| **Workload AT (W)** | 114 ± 49 | 111 ± 56 | 117 ± 53 | 112 ± 56 | 0.814 |
| **VCO₂ AT (mL/min)** | 1321 ± 532 | 1374 ± 656 | 1465 ± 665 | 1208 ± 611 | 0.181 |
| **VE AT (L/min)** | 38.06 ± 14.30 | 36.29 ± 13.83 | 38.91 ± 13.21 | 32.15 ± 11.70 | 0.049 |
| **PetCO₂ AT (mmHg)** | 41.92 ± 4.68 | 41.33 ± 5.03 | 40.67 ± 4.60 | 44.09 ± 5.70[*#] | 0.035 |
| **PetO₂ AT (mmHg)** | 104.8 ± 6.5 | 104.5 ± 5.2 | 106.1 ± 5.9 | 100.6 ± 6.5[*#] | 0.033 |

+ p < 0.05 vs STD; * p < 0.05 vs pNAS; # p < 0.05 vs eOR.

*Abbreviations*: STD: standard conditions; eNAS: exclusively nasal breathing; eOR: exclusively oral breathing; pNAS: with just one blocked nostril; CPET: cardiopulmonary exercise test; FEV1: Forced Expiratory Volume in 1 s; FVC: Forced Vital Capacity; SPB: Systolic Blood Pressure; DBP: Diastolic Blood Pressure; HR: Heart Rate; $VO_2$: Oxygen Intake; $VCO_2$: Carbon dioxide production; VE: Minute Ventilation; RR: Respiratory Rate; TV: Tidal Volume; $SpO_2$: Oxygen saturation; $PetCO_2$: end tidal partial pressure of $CO_2$; $PetO_2$: end tidal partial pressure of $O_2$; AT: Anaerobic Threshold.

**Table 2. CPET values at peak exercise.**

| | STD | pNAS | eOR | eNAS | p |
|---|---|---|---|---|---|
| **SBP peak (mmHg)** | 168±37 | 165±40 | 167±31 | 161±33 | 0.876 |
| **DBP peak (mmHg)** | 88±11 | 83±13 | 88±9 | 83±10 | 0.377 |
| **HR peak (bpm)** | 185±10 | 185±10 | 184±11 | 175±13 | 0.123 |
| **HR peak (% pred)** | 100.1±4.8 | 100.3±4.5 | 99.6±5.2 | 94.7±6.0 | 0.125 |
| **VO$_2$ peak (mL/min)** | 2240±751 | 2184±765 | 2241±800 | 1777±581$^{+*\#}$ | 0.021 |
| **VO$_2$ peak (mL/Kg/min)** | 33.38±6.31 | 32.52±6.40 | 33.30±6.98 | 28.03±5.78$^{+*\#}$ | 0.008 |
| **VO$_2$ peak%** | 0.90±0.15 | 0.88±0.15 | 0.89±0.16 | 0.76±0.17$^{+*\#}$ | 0.005 |
| **VE/VCO$_2$slope** | 26.79±4.27 | 28.61±4.01 | 27.28±4.27 | 24.89±3.51$^{*}$ | 0.021 |
| **VO$_2$/work** | 9.75±0.73 | 9.55±0.52 | 9.50±0.72 | 9.42±1.13 | 0.562 |
| **VCO$_2$ peak (mL/min)** | 2701±860 | 2613±967 | 2688±969 | 2068±729$^{+*\#}$ | 0.008 |
| **VE peak (L/min)** | 88.23±20.63 | 86.93±22.44 | 84.84±20.87 | 55.18±18.25$^{+*\#}$ | <0.001 |
| **RR peak** | 41.33±9.40 | 42.38±9.05 | 40.02±9.00 | 30.43±10.64$^{+*\#}$ | <0.001 |
| **RER** | 1.22±0.08 | 1.20±0.07 | 1.21±0.08 | 1.10±0.10$^{+*\#}$ | 0.027 |
| **TV peak (L/min)** | 2.21±0.65 | 2.11±0.59 | 2.20±0.67 | 1.94±0.64$^{+\#}$ | 0.012 |
| **SpO$_2$ peak** | 96.8±1.3 | 97.4±1.0 | 96.8±1.4 | 97.3±1.8 | 0.54 |
| **PetCO$_2$ peak (mmHg)** | 37.25±5.61 | 35.92±5.63 | 36.92±4.93 | 45.00±7.79$^{+*\#}$ | <0.001 |
| **PetO$_2$ peak (mmHg)** | 115.3±5.2 | 116.0±5.3 | 115.3±4.8 | 105.3±9.8$^{+*\#}$ | 0.007 |
| **BR peak** | 43.04±12.06 | 42.37±13.26 | 45.23±12.59 | 40.23±31.81 | 0.519 |
| **Workload peak (W)** | 207.0±70.2 | 203.7±74.1 | 207.7±75.2 | 184.8±61.0$^{+}$ | 0.036 |
| **Y intercept (L/min)** | 2.42±1.76 | 2.55±3.20 | 3.42±2.08 | 4.12±2.04 | 0.508 |

+ p<0.05 vs STD; * p<0.05 vs pNAS; # p<0.05 vs eOR.

*Abbreviations*: STD: standard conditions; eNAS: exclusively nasal breathing; eOR: exclusively oral breathing; pNAS: with just one blocked nostril; CPET: cardiopulmonary exercise test; SPB: Systolic Blood Pressure; DBP: Diastolic Blood Pressure; HR: Heart Rate; VO$_2$: Oxygen Intake; VCO$_2$: Carbon dioxide production; VE: Minute Ventilation; RR: Respiratory Rate; RER: Respiratory Exchange Ration; TV: Tidal Volume; SpO$_2$: Oxygen saturation; PetCO$_2$: end tidal partial pressure of CO$_2$; PetO$_2$: end tidal partial pressure of O$_2$; BR: Breathing Reserve.

W vs 207±7). At the opposite, peak PetCO$_2$ was higher in eNAS (45.0±7.8 mmHg vs. 37.2±5.6 in STD). No significant differences were noted among the other experimental conditions (pNAS, STD, and eOR) (Table 2). In a similar fashion, peak inspiration and expiration times were significantly augmented in eNAS condition (Fig 2, lower panel). During exercise we showed a significant (p=0.02) reduction in VE/VCO$_2$ slope only in the eNAS breathing vs. pNAS (24.9±3.5 vs 28.6±4.0, respectively) (Table 2). At peak exercise Borg scale for dyspnea, but not for muscular fatigue, in the eNAS condition was significantly higher (Fig 3).

Also comparing the spirometry results, forced expiratory volume (FEV1) and forced vital capacity (FVC) were significantly lower only in eNAS condition (Table 1, Fig 2, upper panel).

In a further analysis considering intermediate iso-watt steps of CPET, we detected a progressive increase in VE impairment along with the workload, with a significant reduction in the eNAS condition already at 75% of the exercise (Fig 4). On the other side, the eOR condition was associated with a different VE behavior at the lower step of the exercise (25%), with higher values compared to all the other conditions. This evidence was paralleled by VO$_2$ values at the same step (Fig 4). These alterations in VE behavior with the mouth or nose plugged are reflected in the kinetics of PetO$_2$ and PetCO$_2$.

## Discussion

Our study demonstrates that a breathing technique involving just nasal respiration (eNAS condition) is related to a significant impairment in exercise performance at CPET, mainly linked to a reduced capability to increase VE during maximal

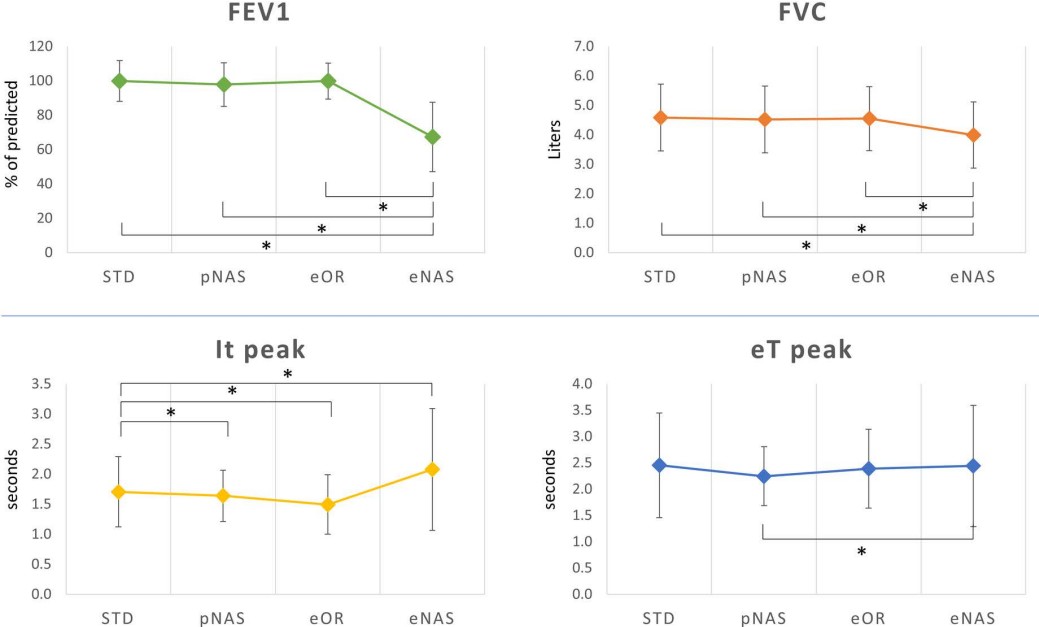

**Fig 2. Spirometry (at rest) and Inspiration/expiration times (peak exercise).** In the upper part of the figure are shown the forced expiratory volume in 1 s (FEV1) and forced vital capacity (FVC) results obtained in the 4 different study conditions at standard spirometry. Lower panel shows the inspiration time (It) and expiration time (Et) at peak exercise at CPET in the 4 different study conditions. *: p < 0.05. Abbreviations: FEV1: Forced Expiratory Volume in 1 s; FVC: Forced Vital Capacity; It: Inspiration time; Et: Expiration Time; STD: standard conditions; eNAS: exclusively nasal breathing; eOR: exclusively oral breathing; pNAS: with just one blocked nostril; CPET: cardiopulmonary exercise test.

exercise. In particular, we show a ~40% reduction in peakVE compared to the normal condition (STD), trigged by a negative effect on both RR and TV. This translates in a reduction in peak workload and namely in key exercise metabolic parameters like peakVO$_2$, and peakVCO$_2$. Despite lower peak RER values, these healthy subjects, compelled by the experimental conditions not to breathe through their mouth, reached their true maximum perceived exertion, as evidenced by markedly elevated respiratory Borg values.

Conversely, a breathing behavior characterized by a partial or total nasal obstruction (pNAS, eOR) was not associated to an objective impairment during a maximal CPET with similar metabolic and respiratory values at peak exercise (Table 2) and translates in a minor effect on the subjective feeling of dyspnea (Fig 3).

Looking to the submaximal phases of the effort, it is interesting how these changes, including the ones recorded in the eNAS breathing, were less pronounced at least until the AT (Table 1), where the eNAS condition was related to a mild reduction in VE and PetO$_2$ and a mild increase in PetCO$_2$. This was also confirmed by a further analysis performed at the same workload steps (25, 50, 75% of the exercise), showing similar "iso-watts" VO$_2$, VCO$_2$ and VE values (Fig 4). In particular, the VE limitation related to eNAS increases in parallel with the workload producing a markedly wider gap starting from 75% of the exercise, and able to affect VO$_2$ values only at peak exercise. These findings are in line with previous data recently published by Eser et al. [24] in a cohort of both healthy and heart failure patients. In their experiment, exercising the subjects at 50% of the maximal exercise in a 5-minutes constant workload testing, they observed very negligible differences in VE with eOR vs eNAS conditions (−0.5 L/min in the healthy subject's group). These data, even in presence of slightly different experimental conditions (maximal incremental CPET vs submaximal workload test) suggests once again, the presence of more pronounced differences at higher exercise stages, as illustrated in the Fig 4 curves. This may suggest even greater differences in the presence of higher exercise intensities, as could occur in highly trained

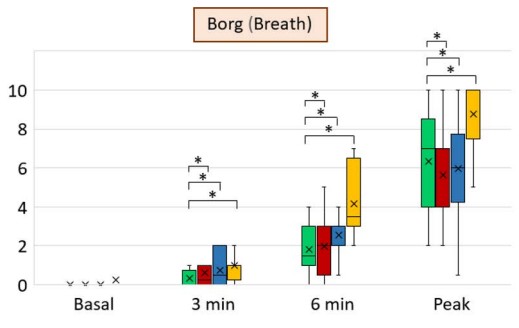

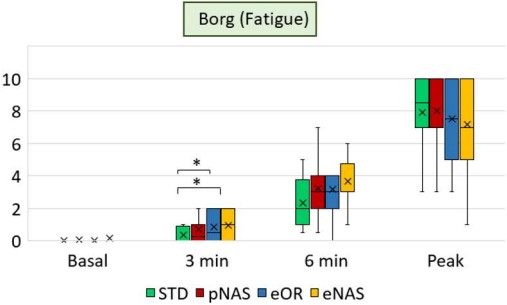

**Fig 3. Borg scale evaluation at rest and during exercise.** Box plot of Borg scale values (perceived breath discomfort in upper part and fatigue in the lower part of the figure) declared by the subjects before the effort (rest), after 3 min of exercise, after 6 min, and at peak exercise at CPET in the 4 different study conditions. *: p<0.05. Abbreviations: STD: standard conditions; eNAS: exclusively nasal breathing; eOR: exclusively oral breathing; pNAS: with just one blocked nostril; CPET: cardiopulmonary exercise test.

athletes, although this has not been investigated in the present study. On opposite, at the lower workload steps (i.e., 25% of the maximal workload), but not at rest, the eOR condition was related to significantly higher VE, PetO$_2$, VO$_2$ values and lower PetCO$_2$ values. These data suggest that in a submaximal exercise situation, before the AT, a forced oral-only (eOR) breathing correlates with an "excess ventilation" relative to the individual's metabolic demands. It is reasonable to assume that, in the absence of the experiment constraints, the subject would predominantly breathe through the nose at this point of the effort. This innate preferred ventilatory pattern is likely related to the lower heat and fluid loss present with nasal respiration. As metabolic demands increase, this difference diminishes until it reverses, with the eOR condition linked to a lower ventilation compared to eNAS at the further exercise steps, just like the other two experimental conditions (STD, pNAS). Taken together, these data demonstrate how, during a submaximal exercise, the nasal respiration is sufficient to maintain the body's metabolic demands, even with a mild self-reported discomfort in the eNAS condition, as shown by a slightly higher Borg already in the first 2 thirds of the exercise (Fig 3). In other words, from a metabolic point of view, the contribution of breathing through the mouth becomes crucial only in the later stages of exercise and particularly at peak. Our data should be read in light of what is already known in literature. At rest and during light cycling exercise the nasal contribution to ventilation is pronounced, with the relative oral contributions increasing substantially as exercise intensity is increased with a relative contribution widely varying among individuals, also according to different races and genders [25]. Two other studied focused on the effect of eOR vs STD conditions during running on peakVO$_2$ [10,26] showing that any additional nasal contribution during oronasal breathing produces no effect on peakVO$_2$ beyond that achievable by oral breathing alone. As a result, available evidence suggests that restricted oral breathing (eOR) and oronasal breathing (STD) are effectively the same in their ability to increase VE and support muscle metabolic demand during maximal exercise. These data are in line with our results. In 1995 Morton et. Al [10] demonstrated a similar drop in peakVO$_2$ and

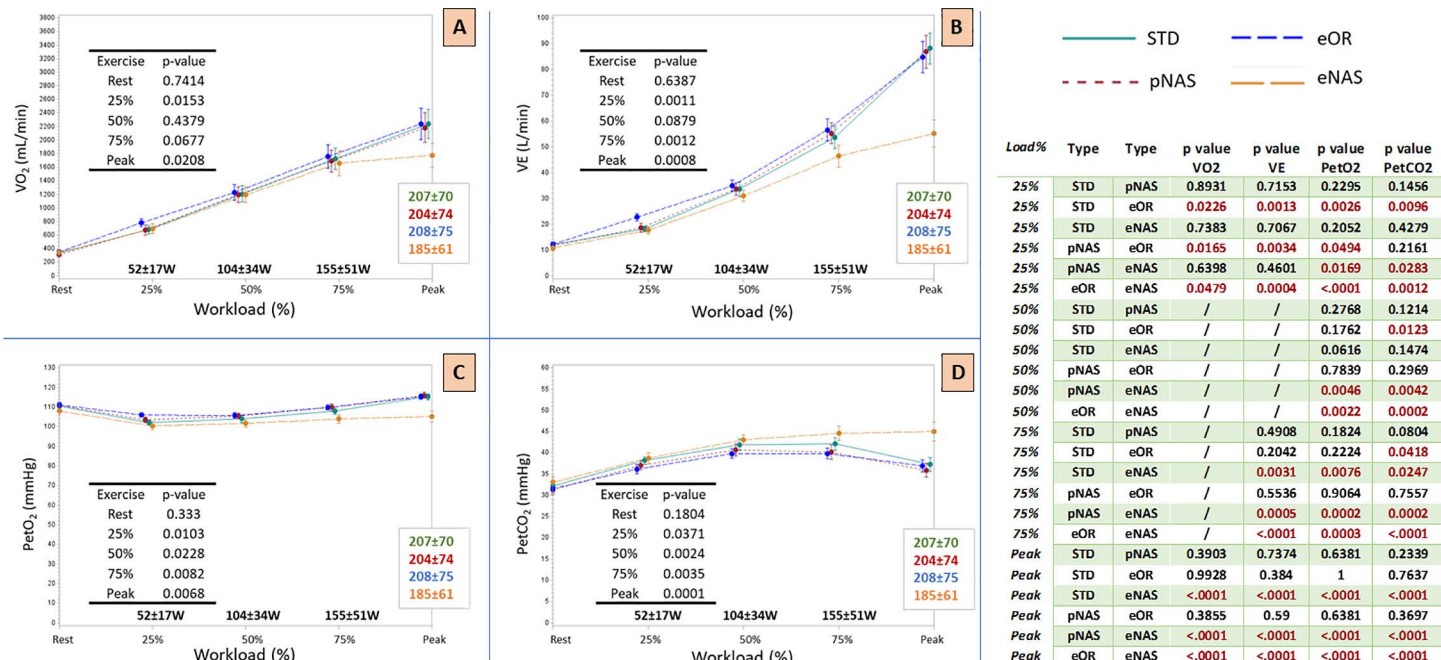

**Fig 4. CPET variables at different exercise steps.** The left side of the figure shows the trend of CPET variables during ramp exercise compared to the percentage of exercise performed. The exercise percentage was calculated with respect to the peak workload reached by each subject during the standard test (STD). 25% workload: 52 ± 17 W; 50% workload: 104 ± 34 W; 75% workload: 155 ± 51 W. Peak workload for STD: 207 ± 70 W; pNAS: 204 ± 74 W; eOR: 208 ± 75 W; eNAS: 185 ± 61 W; eOR: 208 ± 75 W. a) Oxygen uptake (VO$_2$; ml/min). b) Minute ventilation (VE; L/min). c) End tidal partial pressure of O$_2$ (PetO$_2$; mmHg). d) end tidal partial pressure of CO$_2$ (PetCO$_2$; mmHg). The table in the right side of the figure shows the same variables inter-group statistical differences at the different exercise steps. *: p < 0.05. Abbreviations: STD: standard conditions; eNAS: exclusively nasal breathing; eOR: exclusively oral breathing; pNAS: with just one blocked nostril; CPET: cardiopulmonary exercise test; Oxygen uptake (VO$_2$); Minute Ventilation (VE); End tidal partial pressure of CO$_2$ (PetCO$_2$); End tidal partial pressure of O$_2$ (PetO$_2$).

peak workload in healthy subjects, mainly related to a 35% reduction in peak VE. Interestingly, as shown in other previous experiments, the impact of the VE reduction can be mitigated by a specific nasal breathing training, allowing these athletes to reach similar workload and peakVO$_2$ values [6,8]. The authors also suggest that the possible mechanism by which a reduced peak VE can be overcome, consists in a decreased PetO$_2$ and increased PetCO$_2$ in the eNAS condition at a given work level. This phenomenon may result directly from the reduced RR seen when breathing nasally in comparison to breathing orally during exercise, which logically allows more time for diffusion [6,8,10]. This is mirrored by an increased inspiratory and expiratory times (Ti and Te) at peak exercise (Fig 2).

The neutral/minimal effect on submaximal metabolic data also deserves some more comments. Several studies directly examining the effect of nasal versus oral breathing on the ability to complete submaximal endurance exercise (up to 80% of peakVO$_2$) in subjects not specifically accustomed to nasally restricted breathing [7,9,10] suggest that healthy individuals can perform such work without any specific need for adaptation to breathing in a nasally restricted manner.

Furthermore, for the first time, our study demonstrated that also a partial nasal restriction (with just 1 blocked nostril, pNAS) has no significant effect on metabolic and ventilatory variables measured at different exercise phases. The latter situation, artificially recreated in the laboratory, is actually quite common in people's daily lives because it mimics what occurs during colds in which the breath condition accounts only one nostril being able to participate in VE.

These alterations in VE behavior with the mouth or nose plugged are reflected in the kinetics of PetO$_2$ and PetCO$_2$, confirming an excess of ventilation in the eOR condition and vice versa a deficit in the eNAS condition. PetCO$_2$, PetO$_2$,

and VE data confirmed that the eOR condition is related to increased (excessive?) ventilation in the submaximal phases of the exercise, while eNAS produced exactly the opposite effect. Although not directly aimed to compare different voluntary breathing techniques, our study suggests that the *wisest* breathing strategy during an incremental exercise would involve a gradual increase in the oral contribution to VE from the AT until the end of the effort. These preliminary data may act as starting point for further studies exploring the role of different protocols of exercise training involving different breathing techniques aimed to improved exercise performance.

In conclusion, our study demonstrated that in young non-smoker healthy subjects, an exclusively nasal respiration induces significant impairment on peak exercise capacity at CPET due to ventilatory limitation, with only minor effects on metabolic parameters at rest and in submaximal effort. No significant effects were noted in subjects breathing with just one nostril blocked.

## Study limitations

We only enrolled non-smoker subjects, therefore we do not know what happens with their smoker counterparts. However, since this study is focused on the upper airways and not specifically on gas exchange at the pulmonary level (which is the most compromised in chronic obstructive pulmonary disease), it is likely that the effect could still be reproducible, albeit to a different extent.

Our subjects were not trained in different types of breathing before the CPETs. Therefore, these data may not be reproducible in subjects trained to exercise with nose (or mouth) occlusion in the prior weeks, neither with nasal dilators as evidenced in previous studies [27]. Moreover, as suggested by Tong et. al, a reduction in nasal resistance could affect exercise performance [27–29] but this aspect was not considered in the current trial. Similarly, elite athletes may exhibit different behaviours under the same circumstances; future trials conducted on this population may be helpful.

We studied only European subjects. It is possible that in races with different nose and nostrils forms, such as African who show large nose and nostril or Eskimo subjects who have very small nostrils, the effects of the different breathing modalities we tested can vary from what we observed in white European subjects.

Another limitation that should be considered is the small sample size. According the effect of different body size or gender was not analysed. However, our data seem to be very consistent and well aligned with the existing literature on the subject.

## Author contributions

**Conceptualization:** Piergiuseppe Agostoni.

**Data curation:** Massimo Mapelli, Elisabetta Salvioni, Irene Mattavelli, Jeness Campodonico.

**Formal analysis:** Elisabetta Salvioni, Nicolò Capra, Arianna Galotta.

**Investigation:** Massimo Mapelli, Elisabetta Salvioni, Irene Mattavelli, Giulia Grilli, Gabriele Zerboni, Alessandro Nava, Matteo Biroli, Gaia Bellini, Mattia Dall'Asta, Elisabetta Pasini, Antonio De Paola, Ludovica Torzolini, Nicola Mani, Sebastiano Turri.

**Methodology:** Irene Mattavelli.

**Supervision:** Massimo Mapelli, Piergiuseppe Agostoni.

**Validation:** Massimo Mapelli, Elisabetta Salvioni, Irene Mattavelli, Giulia Grilli, Gabriele Zerboni, Alessandro Nava, Arianna Galotta, Gaia Bellini, Mattia Dall'Asta, Elisabetta Pasini, Antonio De Paola, Ludovica Torzolini, Nicola Mani, Sebastiano Turri, Jeness Campodonico, Piergiuseppe Agostoni.

**Visualization:** Elisabetta Salvioni, Irene Mattavelli, Giulia Grilli, Gabriele Zerboni, Alessandro Nava, Arianna Galotta, Gaia Bellini, Mattia Dall'Asta, Elisabetta Pasini, Antonio De Paola, Ludovica Torzolini, Nicola Mani, Sebastiano Turri, Jeness Campodonico, Piergiuseppe Agostoni.

**Writing – original draft:** Massimo Mapelli, Elisabetta Salvioni, Irene Mattavelli, Giulia Grilli, Jeness Campodonico, Piergiuseppe Agostoni.

**Writing – review & editing:** Gabriele Zerboni, Alessandro Nava, Nicolò Capra, Arianna Galotta, Matteo Biroli, Gaia Bellini, Mattia Dall'Asta, Elisabetta Pasini, Antonio De Paola, Ludovica Torzolini, Nicola Mani, Sebastiano Turri.

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
