## [Decision Letter · Decision Letter 0]

PONE-D-24-54440Nasal vs. oral BREATHing WIn Strategies in healthy individuals during cardiorespiratory Exercise testing (BreathWISE)PLOS ONE

Dear Dr. Mapelli,

Thank you for submitting your manuscript to PLOS ONE. After careful consideration, we feel that it has merit but does not fully meet PLOS ONE’s publication criteria as it currently stands. Therefore, we invite you to submit a revised version of the manuscript that addresses the points raised during the review process.

We look forward to receiving your revised manuscript.

Kind regards,

Rodrigo Zacca, Ph.D

Academic Editor

PLOS ONE

Additional Editor Comments:

Thank you for choosing PLOS ONE for the submission of your manuscript.

Your manuscript has been carefully assessed by three experienced reviewers who dedicated their time and effort to provide thorough, constructive, and insightful feedback.

We are grateful for their expertise in ensuring a rigorous review process.

Following this review, they have identified several concerns (MAJOR REVISION) that impact the overall strength and clarity of the study. These include:

- Use of a small and non-representative sample

- Lack of ventilatory flow measurements

- Deficient statistical analyses

- Lack of full data availability

- Inconsistent terminology and methodology details

- Unreferenced claims and missing citations

- Lack of comparison with recent literature

- Unaddressed influence of the nasal cycle

- Missing discussion on the study’s added value

Given these issues, I encourage the authors to carefully review the reviewers’ comments and consider strengthening your study accordingly.

Should you wish to resubmit, a substantially revised version (with tracked changes) addressing these key concerns would be required.

Best Regards

Rodrigo Zacca, Ph.D

Reviewers' comments:

Reviewer's Responses to Questions

**Comments to the Author**

1. Is the manuscript technically sound, and do the data support the conclusions?

Reviewer #1: Yes

Reviewer #2: Yes

Reviewer #3: Partly

2. Has the statistical analysis been performed appropriately and rigorously? 

Reviewer #1: Yes

Reviewer #2: Yes

Reviewer #3: No

3. Have the authors made all data underlying the findings in their manuscript fully available?

Reviewer #1: Yes

Reviewer #2: No

Reviewer #3: Yes

4. Is the manuscript presented in an intelligible fashion and written in standard English?

Reviewer #1: Yes

Reviewer #2: Yes

Reviewer #3: Yes

5. Review Comments to the Author

Reviewer #1: The paper by Mapelli and colleagues addresses a curious and interesting topic in exercise and functional capacity.

The paper is well written, but I would make some minor changes.

- A recent paper by Eser P et al published in Front Physiol 2024 Aug 6;15:1380562 addresses a similar topic: I would discuss differences and added value of your paper compared with data already in the literature.

-Some published work shows the importance of nasal breathing in terms of functional capacity (Tong TK, Effect of nostril dilatation on prolonged all-out intermittent exercise performance. J Sports Med Phys Fitness. 2001; Rappelt L, Front Physiol. 2023 Apr 20;14:1134778): could the fact that you found no differences in exclusively nasal breathing be because the nasal cycle was not considered? I would include among the limitations of the study this consideration.

Reviewer #2: Many thanks for the opportunity to peer review this article. Reviewing the literature, I agree with the author this work adds support to the findings of the other literature highlighted in the discussion. The number of participants in this study is low however this limitation is acknowledged by the author and twelve subjects could be felt to be comparable to the numbers recruited in the other studies referred to in the text i.e. Morton et al. (1995) with 20 subjects, LaComb et al. (2017) with 19 subjects. In brief the study found that a breathing technique which involved just nasal respiration referred to as the eNAS condition caused a significant impairment in exercise performance during cardiopulmonary exercise testing (CPET).

My recommendations are as follows:

Major: The authors do not make available all their data a requirement of the data availability section of the PLOS One submission. I quote: “Authors are required to make all data underlying the findings described fully available, without restriction, and from the time of publication. PLOS allows rare exceptions to address legal and ethical concerns” The authors response to this question is No- some restrictions will apply with Repository of raw data available after acceptance: www.zenodo.org upon request to direzione.scientifica@cardiologicomonzino.it.

Minor: The article does contain some typos and inconsistent language for example the subject is referred to as subjects, participants and in other situations as patients. The term patient particularly should be avoided in the text and the article could benefit from more specifics on the inclusion/exclusion criteria. An aspect I found particularly vague in the methodology is the statement the ‘absence of lateral air leakage was carefully verified as a standard used procedure in CPET laboratories before each test.’ What is this standard used procedure in CPET laboratories? please either reference or describe the methodology. Additionally, the statement all participants signed a written informed consent, and the protocol was approved by the local Ethics Committee (R1925/24 – L2-129) in my opinion should come at the beginning of the methods as signing informed consent form was on their inclusion criteria so it is slightly repetitive. Furthermore, the author states CPET data were analysed breath by breath except peak VO2 analysis (averaged 20 seconds) yet earlier the author states: peak VO2 was calculated as the 30 seconds average of the highest VO2 recorded.

The statement: a maximal or nearly maximal effort was reached, as confirmed by RER>1.05 in all cases. Please reference the ERS statement for criteria of using RER>1.05 for a maximal test. It would also be beneficial to add heart rate data to further support these tests being maximal with a maximal test often being a peak heart rate >100% in adults which can be referenced from the same statement.

(Radtke et al., 2019) RADTKE, T., CROOK, S., KALTSAKAS, G., LOUVARIS, Z., BERTON, D., URQUHART, D. S., KAMPOURAS, A., RABINOVICH, R. A., VERGES, S., KONTOPIDIS, D., BOYD, J., TONIA, T., LANGER, D., DE BRANDT, J., GOËRTZ, Y. M. J., BURTIN, C., SPRUIT, M. A., BRAEKEN, D. C. W., DACHA, S., FRANSSEN, F. M. E., LAVENEZIANA, P., EBER, E., TROOSTERS, T., NEDER, J. A., PUHAN, M. A., CASABURI, R., VOGIATZIS, I. & HEBESTREIT, H. 2019. ERS statement on standardisation of cardiopulmonary exercise testing in chronic lung diseases. Eur Respir Rev, 28.

Finally, within the discussion there is the statement within the limitations section suggesting data may not be reproducible in subjects trained to exercise with nose (or Mouth) occlusion in the prior weeks as evidenced in previous studies. Please reference/refer to the studies to support this statement.

Reviewer #3: The article entitled "Nasal vs. Oral BREATHing Win Strategies in Healthy Individuals during Cardiorespiratory Exercise Testing (BREATHWISE)", is concerned with the alterations in ventilation and metabolism that occur in response to the partial or complete obstruction either of the nasal or oral airways during an exercise test. The main point of this article is the intervention on the nose and the mouth of humans to change the ventilation pattern during incremental exercise tests. This intervention aims to alter normal breathing during exercise.

The interest of this paper is to deal with the time to exchange respiratory gases, induced by higher flow resistance through the nasal pathway, and its possible application to air pathway features in pathology such as COPD or yoga or tai-chi strategies. Due to the characteristics of the CPETs carried out and their results data shown scarcely apply to athletic performance

Major concerns

They recruited six men and six women, a small sample size which makes it difficult to extrapolate the data to the general population. The data from the exercise tests show that the subjects were not in good physical condition (VO2 and VE), what left real maximal intensities unexplored. Furthermore, as occlusion of either the nose or mouth changes airflow resistance, a measurement of this variable at rest and during exercise is missing. As the nasal pathway represents two-thirds of the total resistance of the upper airways, occlusion of the mouth obviously increases upper airway resistances and consequently changes the ventilation pattern. Again there is a lack of recording of airflow characteristics such as ventilatory peak flow (i.e.TV/ti, TV/te), average flow, and tidal volume time constant. The authors do not report the predetermination of the sample size. From a statistical view, authors show mean ± standard deviation data but they do not explain if mean and SD correspond to each subject during every condition or, contrariwise, the whole group of experimental subjects. Since they examined 6 men and 6 women it is important to analyze women separately from men otherwise the smaller ventilatory volumes and the higher ventilatory frequency of women can introduce a bias in the results. Statistical data also must include the confidence interval for every variable and in every category. The above weakens the structure of this paper

Minor concerns

Title

#1-In the title appear Win and BreathWISE, both terms are not explained in the rest of the article. At least, Breath WISE should be explained as a structured approach to breathing techniques aimed at improving lung function, reducing stress and enhancing overall well-being. Contrariwise, both terms should be changed for “breathing”.

#2- It is not well described whether the specific strategies lead to an improvement in exercise performance or whether they have been directed towards breathing therapies.

#3- In the Abstract, the sentences “Twelve healthy subjects (28.6±5.2 y, 50% males) performed the 4 CPETs within one month. Variables were analysed at rest, at anaerobic threshold (AT), at intermediate exercise steps, and at peak” that appear in Results, should be move to Methods in Abstract.

#4- In the fifth line of the Results paragraph, in the Abstract, authors write “Moreover, peak inspiration and peak expiration time were augmented, while forced expiratory volume and vital capacity at rest were reduced. On opposite, only minor differences were detected at rest o AT”. If the variables forced expiratory volume and vital capacity were reduced at rest, what is the meaning of “On opposite, only minor differences were detected at rest or AT”? The authors should consider explaining or deleting this sentence. Also the terms peak inspiration and peak expiratory time might be changed for “inspiratory and expiratory time to peak” for the sake of clarity.

#5- In Methods, the experimental subjects performed exercise in a cycle ergometer in which the subjects were allowed to see “rpm” (revolution per minute) but author did not tell if the rpm value was the same for the group or was different for each subjects or was fixed during the four conditions. It ought be clarified.

In statistical analysis the authors write that when the distribution of data belonged to a normal population they used mean±SD and when data belonged to a non-normal distribution they used median and interquartile. However normal and non-normal distributions were not assigned to the variables analysed in the article. It seems a stock phrase for statistic more than actual analysis.

#6- RESULTS: The values of VO2 peak and VCO2 peak reveal a low training status of the subjects. Attending to VO2 and VCO2 peaks the tests should be classified as submaximal. Also HR peaks did not reach the maximum corresponding to the average age of the sample. The workload peak around 200 W is not a value of athletes. By other side, it is known that women show higher HR than men what may introduce a bias in the statiscal analysis of this variable.

#7 Figure 2. Both panels are lacking the units in the ordinate axis. In FEV1 panel the ordinate values seem to be in percentage and FVC in litres; a Tiffenau index would be welcome. In the lower panel the ordinate scales should be in the same range to allow values to be compared. Provided the values of iT and eT (units?) in STD, pNAS and eOR conditions is difficult to believe in the significant differences among them, given the small sample and the standard deviation magnitudes. Further, in addition to the longer time for expiration as compared with inspiration, does not seem to there be significant differences in the eT values, including the eT peak value.

Provided the ventilatory frequency (30.43 in eNAS vs 40 in eOR) and the PetCO2 peak (45 in eNAS vs 36.92 in eOR) of the eNAS condition (Table 2), it is not easy to explain how an increase in the end tidal CO2 is not correlated with higher ventilatory frequency in eNAS. It is well known than increase CO2 undergo an increase in ventilatory frequency. An eT longer than iT can help to explain the higher PetCO2 value in eNAS condition but not the change in ventilator frequency. May be these data deserve a better. explanation.

#8- Figure 4 The statistical differences applied to each workload step have no sense because increasing workload between steps (like Bruce Protocol) is mandatory. Moreover, this does not seem the main objective of the article but the different ventilation pattern among categories.. However, does there not be differences between each condition during the three first workload steps (in X axis). At least, differences between the eNAS and the other conditions in the last step of the CPET are clear. Again, the PetCO2 and the VE values deserve a better explanation.

6. PLOS authors have the option to publish the peer review history of their article (what does this mean? ). If published, this will include your full peer review and any attached files.

**Do you want your identity to be public for this peer review?** For information about this choice, including consent withdrawal, please see our Privacy Policy .

Reviewer #1: No

Reviewer #2: No

Reviewer #3: No

---

## [Author Response · Author response to Decision Letter 1]

14 Mar 2025

Response to the Reviewers

Manuscript: Nasal vs. oral BREATHing WIn Strategies in healthy individuals during cardiorespiratory Exercise testing (BreathWISE)

PLOS ONE (PONE-D-24-54440)

Editor’s comment

Thank you for choosing PLOS ONE for the submission of your manuscript.

Your manuscript has been carefully assessed by three experienced reviewers who dedicated their time and effort to provide thorough, constructive, and insightful feedback.

We are grateful for their expertise in ensuring a rigorous review process.

Following this review, they have identified several concerns (MAJOR REVISION) that impact the overall strength and clarity of the study

We thank the Editor for the possibility to revise our manuscript and for the positive comments received.

We acknowledge the effort of the 3 Reviewers and are very grateful for their positive and constructive comments. We have prepared a step by step reply and a new version of the paper in accordance to the comments received. We hope that they are satisfactorily and we believe that the manuscript is now improved.

Reviewer #1: The paper by Mapelli and colleagues addresses a curious and interesting topic in exercise and functional capacity.

The paper is well written, but I would make some minor changes.

We thank the Reviewer#1 for his positive feedback and for the suggestions provided. The manuscript has been modified accordingly (see below the point-by-point answers).

- A recent paper by Eser P et al published in Front Physiol 2024 Aug 6;15:1380562 addresses a similar topic: I would discuss differences and added value of your paper compared with data already in the literature.

Thank you for this important comment. We were not aware of this manuscript when we submitted the manuscript but we read it few weeks later. It is also very interesting and address similar topics. However, we believe that the 2 cohort and the methods differ significantly. In particular, we enrolled only healthy subjects and not patients and – more importantly – they exercised patient at a submaximal level (50% of the peak) while we performed maximal CPET. Indeed, comparing our data at 50% of the exercise (see Figure 4), we also did not find significant differences in term of VO2 and VE, as expressed in the “Young control subjects” by Eser et al. (see their Table 3, last columns, in which they also found a very minimal difference – 0.5 L/min – in VE). According to your comment we have expanded the discussion also adding the citation. Specifically, we now report “These findings are in line with previous data recently published by Eser et al. 24 in a cohort of both healthy and heart failure patients. In their experiment, exercising the subjects at 50% of the maximal exercise in a 5-minutes constant workload testing, they observed very negligible differences in VE with eOR vs eNAS conditions (-0.5 L/min in the healthy subject’s group). These data, even in presence of slightly different experimental conditions (maximal incremental CPET vs submaximal workload test) suggests once again, the presence of more pronounced differences at higher exercise stages, as illustrated in the Figure 4 curves. This may suggest even greater differences in the presence of higher exercise intensities, as could occur in highly trained athletes, although this has not been investigated in the present study.”

-Some published work shows the importance of nasal breathing in terms of functional capacity (Tong TK, Effect of nostril dilatation on prolonged all-out intermittent exercise performance. J Sports Med Phys Fitness. 2001; Rappelt L, Front Physiol. 2023 Apr 20;14:1134778): could the fact that you found no differences in exclusively nasal breathing be because the nasal cycle was not considered? I would include among the limitations of the study this consideration.

Thank you for this comment. We have added a dedicated sentence to the limitation section as suggested: “Although not directly aimed to compare different voluntary breathing techniques, our study suggests that the wisest breathing strategy during an incremental exercise would involve a gradual increase in the oral contribution to VE from the AT until the end of the effort. These preliminary data may act as starting point for further studies exploring the role of different protocols of exercise training involving different breathing techniques aimed to improved exercise performance.”

Reviewer #2: Many thanks for the opportunity to peer review this article. Reviewing the literature, I agree with the author this work adds support to the findings of the other literature highlighted in the discussion. The number of participants in this study is low however this limitation is acknowledged by the author and twelve subjects could be felt to be comparable to the numbers recruited in the other studies referred to in the text i.e. Morton et al. (1995) with 20 subjects, LaComb et al. (2017) with 19 subjects. In brief the study found that a breathing technique which involved just nasal respiration referred to as the eNAS condition caused a significant impairment in exercise performance during cardiopulmonary exercise testing (CPET).

We really thank the Reviewer#2 for his nice feedback and for his comments. The manuscript has been modified accordingly (see below the point-by-point answers). We believe that following your suggestions it significantly improved.

My recommendations are as follows:

Major: The authors do not make available all their data a requirement of the data availability section of the PLOS One submission. I quote: “Authors are required to make all data underlying the findings described fully available, without restriction, and from the time of publication. PLOS allows rare exceptions to address legal and ethical concerns” The authors response to this question is No- some restrictions will apply with Repository of raw data available after acceptance: www.zenodo.org upon request to direzione.scientifica@cardiologicomonzino.it.

We will change the access to the repository as “without any restriction” as requested.

Minor: The article does contain some typos and inconsistent language for example the subject is referred to as subjects, participants and in other situations as patients. The term patient particularly should be avoided in the text and the article could benefit from more specifics on the inclusion/exclusion criteria.

You are right and we are sorry. Indeed, the term “patients” was present mostly in the methods section since usually we performed CPET in HF patients. We have fixed the text leaving only the terms “subjects” or “participants” that are more appropriate. Thank you

An aspect I found particularly vague in the methodology is the statement the ‘absence of lateral air leakage was carefully verified as a standard used procedure in CPET laboratories before each test.’ What is this standard used procedure in CPET laboratories? please either reference or describe the methodology.

Thank you. According to your comment, we have modified the text and added a specific reference: “The absence of lateral air leakage was carefully verified as a standard used procedure in CPET laboratories before each test by, completely blocking the ventilation valve of the spirometry mask with the palm of the hand as previously described 11.”

Additionally, the statement all participants signed a written informed consent, and the protocol was approved by the local Ethics Committee (R1925/24 – L2-129) in my opinion should come at the beginning of the methods as signing informed consent form was on their inclusion criteria so it is slightly repetitive.

Done as requested. Thank you

Furthermore, the author states CPET data were analyzed breath by breath except peak VO2 analysis (averaged 20 seconds) yet earlier the author states: peak VO2 was calculated as the 30 seconds average of the highest VO2 recorded.

You are right, the right value is 30 seconds. 20 was written by mistake. We fixed it in the text. Thank you.

The statement: a maximal or nearly maximal effort was reached, as confirmed by RER>1.05 in all cases. Please reference the ERS statement for criteria of using RER>1.05 for a maximal test.

Thank you. We have added it in the methods section

It would also be beneficial to add heart rate data to further support these tests being maximal with a maximal test often being a peak heart rate >100% in adults which can be referenced from the same statement.

(Radtke et al., 2019) RADTKE, T., CROOK, S., KALTSAKAS, G., LOUVARIS, Z., BERTON, D., URQUHART, D. S., KAMPOURAS, A., RABINOVICH, R. A., VERGES, S., KONTOPIDIS, D., BOYD, J., TONIA, T., LANGER, D., DE BRANDT, J., GOËRTZ, Y. M. J., BURTIN, C., SPRUIT, M. A., BRAEKEN, D. C. W., DACHA, S., FRANSSEN, F. M. E., LAVENEZIANA, P., EBER, E., TROOSTERS, T., NEDER, J. A., PUHAN, M. A., CASABURI, R., VOGIATZIS, I. & HEBESTREIT, H. 2019. ERS statement on standardisation of cardiopulmonary exercise testing in chronic lung diseases. Eur Respir Rev, 28.

As suggested, we have added the percent of predicted value of HR in table 2, and modified the method section accordingly: “A maximal RER during exercise exceeding 1.05 was considered an indicator of a maximal effort 17. Similarly, a maximal peak HR above 100% was also used as a supportive criterion 17-19.” As you can see, all the subject reached at least the 85% of their predicted HR (a more conservative cut-off used to indicate maximal exercise in the literature), but with average values far more than 95%. We have modified the text accordingly.

Finally, within the discussion there is the statement within the limitations section suggesting data may not be reproducible in subjects trained to exercise with nose (or Mouth) occlusion in the prior weeks as evidenced in previous studies. Please reference/refer to the studies to support this statement.

Thank you, added as requested, also according to the other Reviewer’s comments. We have also re-phrased that part of the text for clarity. Specifically:” Therefore, these data may not be reproducible in subjects trained to exercise with nose (or mouth) occlusion in the prior weeks, neither with nasal dilators as evidenced in previous studies27. Moreover, as suggested by Tong et. al, a reduction in nasal resistance could affect exercise performance 27-29 but this aspect was not considered in the current trial. Similarly, elite athletes may exhibit different behaviours under the same circumstances; future trials conducted on this population may be helpful.”

Reviewer #3: The article entitled "Nasal vs. Oral BREATHing Win Strategies in Healthy Individuals during Cardiorespiratory Exercise Testing (BREATHWISE)", is concerned with the alterations in ventilation and metabolism that occur in response to the partial or complete obstruction either of the nasal or oral airways during an exercise test. The main point of this article is the intervention on the nose and the mouth of humans to change the ventilation pattern during incremental exercise tests. This intervention aims to alter normal breathing during exercise.

The interest of this paper is to deal with the time to exchange respiratory gases, induced by higher flow resistance through the nasal pathway, and its possible application to air pathway features in pathology such as COPD or yoga or tai-chi strategies. Due to the characteristics of the CPETs carried out and their results data shown scarcely apply to athletic performance

Thank you for your positive feedback and your insightful comments. We agree that our results are not entirely applicable to athletic subjects, since we enrolled an unselected population of healthy individuals. Further studies on athletes would be desirable, and this is now added to the limitation section.

Major concerns

They recruited six men and six women, a small sample size which makes it difficult to extrapolate the data to the general population. The data from the exercise tests show that the subjects were not in good physical condition (VO2 and VE), what left real maximal intensities unexplored.

Thank you for your comment. We agree that the physical condition of our individuals is not perfect, but definitely within a normal range (average VO2 in the STD condition is 33.4 ml/kg/min, corresponding to 90% of the predicted). However, as mentioned before, our aim was not to describe the effects of different respiration strategies in athletes, but only in the general healthy population. Following your suggestions, the text has been modified in the introduction as well as in the discussion and limitations sections. Specifically: “This may suggest even greater differences in the presence of higher exercise intensities, as could occur in highly trained athletes, although this has not been investigated in the present study.” ….and …”elite athletes may exhibit different behaviors under the same circumstances; future trials conducted on this population may be helpful”

Furthermore, as occlusion of either the nose or mouth changes airflow resistance, a measurement of this variable at rest and during exercise is missing. As the nasal pathway represents two-thirds of the total resistance of the upper airways, occlusion of the mouth obviously increases upper airway resistances and consequently changes the ventilation pattern. Again, there is a lack of recording of airflow characteristics such as ventilatory peak flow (i.e.TV/ti, TV/te), average flow, and tidal volume time constant.

I totally agree with your comment that eNAS have a higher airflow resistance and this explains the reduced ventilation observed in this condition (please see reply to your last comment below) and the more prolonged Ti. It is also true that we did not report some ventilation measurements such as TV/Ti and TV/te. However, TV and inspiratory and expiratory time were reported independently (see figure 2). I think that, to avoid overwhelming the reader with too many measurements there is no need to add these values. Please, note that the exercise protocol was a ramp (progressively increasing workload) so that it is not possible to calculate the time constant.

The authors do not report the predetermination of the sample size.

Sample size evaluation has been added to the statistical paragraph as you requested. Specifically: “Sample size determination: With a sample of 12 subjects we planned to identify an effect size of 1.2, with a power of 80%, considering an alpha=0.05 and an ANOVA for repeated measures (4 measurements).”

From a statistical view, authors show mean ± standard deviation data but they do not explain if mean and SD correspond to each subject during every condition or, contrariwise, the whole group of experimental subjects. Since they examined 6 men and 6 women it is important to analyze women separately from men otherwise the smaller ventilatory volumes and the higher ventilatory frequency of women can introduce a bias in the results. Statistical data also must include the confidence interval for every variable and in every category. The above weakens the structure of this paper.

Data are expressed as mean and SD as all tested variables in our study were normally distributed. Reported means refers to all patients in each condition. I agree that women and men have different ventilatory volumes, but as all the analysis were paired (within subject) we believe that results obtained are still meaningful. However, you are completely right and further studies with greater number of subjects would be needed to appreciate sex related differences.

As we completely agree with your comment we already included this aspect as a limitation in the text.

Minor concerns

Title

#1-In the title appear Win and BreathWISE, both terms are not explained in the rest of the article. At least, Breath WISE should be explained as a structured approach to breathing techniques aimed at improving lung function, reducing stress and enhancing overall well-being. Contrariwise, both terms should be changed for “breathing”.

Thank you for your comment. The BreathWISE was chosen as an acronym (see the capital letters in the title). Following your observation, to make it clearer, and to better summarize out finding, the following sentence has been added in the discussion:

“Although not directly aimed to co

---

## [Decision Letter · Decision Letter 1]

PONE-D-24-54440R1Nasal vs. oral BREATHing WIn Strategies in healthy individuals during cardiorespiratory Exercise testing (BreathWISE)PLOS ONE

Dear Dr. Mapelli,

Thank you for submitting your manuscript to PLOS ONE. After careful consideration, we feel that it has merit but does not fully meet PLOS ONE’s publication criteria as it currently stands. Therefore, we invite you to submit a revised version of the manuscript that addresses the points raised during the review process.

We look forward to receiving your revised manuscript.

Kind regards,

Rodrigo Zacca, Ph.D

Academic Editor

PLOS ONE

Additional Editor Comments:

The authors should report the test of sphericity (and apply corrections if violated), the F-statistic (F), degrees of freedom (df), and effect size (partial eta squared, η²p), along with an interpretation of its magnitude. Additionally, the mean differences (both absolute and percentage) between each condition should be reported, as well as the 95% confidence intervals for all post-hoc comparisons. Furthermore, the quality of the graphs in Figure 2 is very low. I strongly recommend using GraphPad Prism to improve the quality and clarity of the figures. After that, please edit the remaining sessions accordingly.

Reviewers' comments:

Reviewer's Responses to Questions

**Comments to the Author**

1. If the authors have adequately addressed your comments raised in a previous round of review and you feel that this manuscript is now acceptable for publication, you may indicate that here to bypass the “Comments to the Author” section, enter your conflict of interest statement in the “Confidential to Editor” section, and submit your "Accept" recommendation.

Reviewer #1: All comments have been addressed

Reviewer #2: All comments have been addressed

2. Is the manuscript technically sound, and do the data support the conclusions?

Reviewer #1: Yes

Reviewer #2: Yes

3. Has the statistical analysis been performed appropriately and rigorously? 

Reviewer #1: Yes

Reviewer #2: Yes

4. Have the authors made all data underlying the findings in their manuscript fully available?

Reviewer #1: Yes

Reviewer #2: Yes

5. Is the manuscript presented in an intelligible fashion and written in standard English?

Reviewer #1: Yes

Reviewer #2: Yes

6. Review Comments to the Author

Reviewer #1: Thank you for the timely and accurate response to the comments raised, to which the authors have responded appropriately.

I have no further comments to make.

Reviewer #2: (No Response)

7. PLOS authors have the option to publish the peer review history of their article (what does this mean? ). If published, this will include your full peer review and any attached files.

**Do you want your identity to be public for this peer review?** For information about this choice, including consent withdrawal, please see our Privacy Policy .

Reviewer #1: No

Reviewer #2: No

---

## [Author Response · Author response to Decision Letter 2]

26 May 2025

Additional Editor Comments:

The authors should report the test of sphericity (and apply corrections if violated), the F-statistic (F), degrees of freedom (df), and effect size (partial eta squared, η²p), along with an interpretation of its magnitude. Additionally, the mean differences (both absolute and percentage) between each condition should be reported, as well as the 95% confidence intervals for all post-hoc comparisons. Furthermore, the quality of the graphs in Figure 2 is very low. I strongly recommend using GraphPad Prism to improve the quality and clarity of the figures. After that, please edit the remaining sessions accordingly.

Thank you for the suggestion. We have addressed your comment by including both the multivariate analysis (Wilks’ Lambda) to assess the overall effect of time across dependent variables, and the detailed univariate analyses for each variable separately.

Specifically, in the univariate analyses, we have reported:

• The Mauchly’s test of sphericity,

• Greenhouse-Geisser correction in case of violation,

• The F-statistic, corrected degrees of freedom, and p-values,

• The partial eta squared (η²p) as a measure of effect size.

However, to maintain clarity and focus in the main manuscript, we have chosen not to include these detailed statistical outputs in the main text. As requested, we have nonetheless provided an Excel file containing all the analyses, including absolute and percentage deltas with their respective 95% confidence intervals. These tables are quite complex and dense, and while they do not alter the overall message or conclusions of the paper, we are happy to provide them to the Editor for full transparency.

Should the Editor deem it necessary to include some or all of these data in the publication, we would be glad to provide them as supplemental material at their discretion.

Regarding Figure 2, unfortunately the version embedded in the PDF is of low resolution; however, a high-resolution version is available through the download link provided in the file.

---

## [Editor Report · Decision Letter 2]

Nasal vs. oral BREATHing WIn Strategies in healthy individuals during cardiorespiratory Exercise testing (BreathWISE)

PONE-D-24-54440R2

Dear Dr. Mapelli,

On behalf of PLOS ONE, I am pleased to inform you that your manuscript entitled "Nasal vs. oral BREATHing WIn Strategies in healthy individuals during cardiorespiratory Exercise testing (BreathWISE)" has been formally accepted for publication in PLOS One. The manuscript was evaluated through a rigorous peer-review process by independent experts in the field. We extend our sincere appreciation to the reviewers for their thoughtful and constructive feedback, which played a valuable role in shaping the final version of the article. Thank you for submitting your work to PLOS ONE and for contributing to the advancement of open-access science.

Kind regards,

Rodrigo Zacca, Ph.D

Academic Editor

PLOS ONE

Additional Editor Comments (optional):

Dear Dr. Massimo Mapelli,

On behalf of PLOS ONE, I am pleased to inform you that your manuscript entitled "Nasal vs. oral BREATHing WIn Strategies in healthy individuals during cardiorespiratory Exercise testing (BreathWISE)" has been formally accepted for publication in PLOS One. The manuscript was evaluated through a rigorous peer-review process by independent experts in the field. We extend our sincere appreciation to the reviewers for their thoughtful and constructive feedback, which played a valuable role in shaping the final version of the article. Thank you for submitting your work to PLOS ONE and for contributing to the advancement of open-access science.

Sincerely,

Rodrigo Zacca, Ph.D

Academic Editor

PLOS ONE
---

## [Editor Report · Acceptance letter]

PONE-D-24-54440R2

PLOS ONE

Dear Dr. Mapelli,

I'm pleased to inform you that your manuscript has been deemed suitable for publication in PLOS ONE. Congratulations! Your manuscript is now being handed over to our production team.

Kind regards,

on behalf of

Dr. Rodrigo Zacca

Academic Editor

PLOS ONE